# Common Drug Pipelines for the Treatment of Diabetic Nephropathy and Hepatopathy: Can We Kill Two Birds with One Stone?

**DOI:** 10.3390/ijms21144939

**Published:** 2020-07-13

**Authors:** Yoshio Sumida, Masashi Yoneda, Hidenori Toyoda, Satoshi Yasuda, Toshifumi Tada, Hideki Hayashi, Yoichi Nishigaki, Yusuke Suzuki, Takafumi Naiki, Asahiro Morishita, Hiroshi Tobita, Shuichi Sato, Naoto Kawabe, Shinya Fukunishi, Tadashi Ikegami, Takaomi Kessoku, Yuji Ogawa, Yasushi Honda, Takashi Nakahara, Kensuke Munekage, Tsunehiro Ochi, Koji Sawada, Atsushi Takahashi, Taeang Arai, Tomomi Kogiso, Satoshi Kimoto, Kengo Tomita, Kazuo Notsumata, Michihiro Nonaka, Kazuhito Kawata, Taro Takami, Takashi Kumada, Eiichi Tomita, Takeshi Okanoue, Atsushi Nakajima

**Affiliations:** 1Division of Hepatology and Pancreatology, Department of Internal Medicine, Aichi Medical University, Nagakute, Aichi 480-1195, Japan; yoneda@aichi-med-u.ac.jp (M.Y.); kimoto.satoshi.146@mail.aichi-med-u.ac.jp (S.K.); 2Department of Gastroenterology, Ogaki Municipal Hospital, Ogaki, Gifu 503-8502, Japan; hmtoyoda@spice.ocn.ne.jp (H.T.); satoshi.yasuda.1982@gmail.com (S.Y.); takashi.kumada@gmail.com (T.K.); 3Department of Hepatology, Himeji Redcross Hospital, Himeji, Hyogo 670-8540, Japan; tadat0627@gmail.com; 4Department of Gastroenterology, Gifu Municipal Hospital, Gifu 500-8513, Japan; hide-hayashi@umin.ac.jp (H.H.); something@beach.ocn.ne.jp (Y.N.); usukesuzuki0915@yahoo.co.jp (Y.S.); naiki-gif@umin.org (T.N.); etomita_jp@yahoo.co.jp (E.T.); 5Department of Gastroenterology and Neurology, Faculty of Medicine, Kagawa University, Kagawa 761-0793, Japan; asahiro@med.kagawa-u.ac.jp; 6Department of Gastroenterology and Hepatology, Shimane University Faculty of Medicine, Izumo, Shimane 693-8501, Japan; ht1020@med.shimane-u.ac.jp; 7Department of Internal Medicine, Izumo City General Medical Center, Izumo, Shimane 691-0003, Japan; bbsato@med.shimane-u.ac.jp; 8Department of Liver, Biliary Tract and Pancreas Diseases, Fujita Health University School of Medicine, Aichi 470-1192, Japan; naoto-kawabe@hotmail.co.jp; 9Premier Development Research of Medicine, Osaka Medical College, Osaka 569-8686, Japan; in2104@osaka-med.ac.jp; 10Department of Gastroenterology, Tokyo Medical University Ibaraki Medical Center, Ibaraki 300-0395, Japan; ikegamit@tokyo-med.ac.jp; 11Department of Gastroenterology and Hepatology, Yokohama City University Graduate School of Medicine, Yohokama, Kanagawa 236-0004, Japan; takaomi0027@gmeil.com (T.K.); yuji.ogawa01@gmail.com (Y.O.); y-honda@umin.ac.jp (Y.H); nakajima-tky@umin.ac.jp (A.N.); 12Department of Gastroenterology and Metabolism, Graduate School of Biomedical and Health Sciences, Hiroshima University, Hiroshima 734-8551, Japan; nakahara@hiroshima-u.ac.jp; 13Department of Gastroenterology and Hepatology, Kochi Medical School, Kochi 780-8505, Japan; jm-k.munekage@kochi-u.ac.jp (K.M.); tsunehiroochi@kochi-u.ac.jp (T.O.); 14Department of Medicine, Division of Gastroenterology and Hematology/Oncology, Asahikawa Medical University, Asahikawa 078-8510, Japan; k-sawada@asahikawa-med.ac.jp; 15Department of Gastroenterology, Fukushima Medical University School of Medicine, Fukushima 960-1295, Japan; junior@fmu.ac.jp; 16Division of Gastroenterology and Hepatology, Nippon Medical School, Tokyo 113-8602, Japan; taeangpark@yahoo.co.jp; 17Department of Internal Medicine, Institute of Gastroenterology, Tokyo Women’s Medical University, Tokyo 162-8266, Japan; kogiso.tomomi@twmu.ac.jp; 18Department of Internal Medicine, National Defense Medical College, Tokorozawa, Saitama 359-8513, Japan; kengo@ndmc.ac.jp; 19Department of General Internal Medicine, Fukui-ken Saiseikai Hospital, Fukui 918-8503, Japan; notsumata.kazuo6017@fukui.saiseikai.or.jp; 20Department of Gastroenterology and Hepatology, JA Hiroshima General Hospital, Hiroshima 738-8503, Japan; syoukakinaika@hirobyo.jp; 21Hepatology Division, Department of Internal Medicine II, Hamamatsu University School of Medicine, Hamamatsu, Shizuoka 431-3192, Japan; kawata@hama-med.ac.jp; 22Department of Gastroenterology and Hepatology, Yamaguchi University, Ube, Yamaguchi 755-8505, Japan; t-takami@yamaguchi-u.ac.jp; 23Hepatology Center, Saiseikai Suita Hospital, Suita, Osaka 564-0013, Japan; okanoue@suita.saiseikai.or.jp

**Keywords:** diabetic nephropathy, diabetic hepatopathy, chronic kidney disease, glucagon-like peptide 1, peroxisome proliferator-activated receptor, sodium–glucose cotransporter 2

## Abstract

Type 2 diabetes (T2D) is associated with diabetic nephropathy as well as nonalcoholic steatohepatitis (NASH), which can be called “diabetic hepatopathy or diabetic liver disease”. NASH, a severe form of nonalcoholic fatty disease (NAFLD), can sometimes progress to cirrhosis, hepatocellular carcinoma and hepatic failure. T2D patients are at higher risk for liver-related mortality compared with the nondiabetic population. NAFLD is closely associated with chronic kidney disease (CKD) or diabetic nephropathy according to cross-sectional and longitudinal studies. Simultaneous kidney liver transplantation (SKLT) is dramatically increasing in the United States, because NASH-related cirrhosis often complicates end-stage renal disease. Growing evidence suggests that NAFLD and CKD share common pathogenetic mechanisms and potential therapeutic targets. Glucagon-like peptide 1 (GLP-1) receptor agonists and sodium–glucose cotransporter 2 (SGLT2) inhibitors are expected to ameliorate NASH and diabetic nephropathy/CKD. There are no approved therapies for NASH, but a variety of drug pipelines are now under development. Several agents of them can also ameliorate diabetic nephropathy/CKD, including peroxisome proliferator-activated receptors agonists, apoptosis signaling kinase 1 inhibitor, nuclear factor-erythroid-2-related factor 2 activator, C-C chemokine receptor types 2/5 antagonist and nonsteroidal mineral corticoid receptor antagonist. This review focuses on common drug pipelines in the treatment of diabetic nephropathy and hepatopathy.

## 1. Introduction

Nonalcoholic fatty liver disease (NAFLD), which is a hepatic manifestation of metabolic syndrome, is the most common chronic liver disease. Worldwide, 25% of the adult population is now suffering from NAFLD [1,2]. Nonalcoholic steatohepatitis (NASH), which is defined by hepatic steatosis with inflammation and ballooning, can progress to cirrhosis, liver failure and hepatocellular carcinoma (HCC). The incidence of NASH has risen due to the increased prevalence of obesity, metabolic syndrome and type 2 diabetes (T2D). In the United States (US), NASH has become the leading cause of liver transplantation [3]. T2D is closely associated with NASH incidence and fibrosis progression. In Japan, liver-related disease is the third leading cause of mortality (9.3%) in T2D according to a nationwide survey (2001–2010) [4]. T2D patients are at higher risk for the development of or mortality from HCC [5,6]. Thus, NASH can be called “diabetic hepatopathy or diabetic liver disease (DLD)” [7]. Hepatic fibrosis is the most significant determinant of overall mortality and liver-related mortality in NAFLD [8]. The estimated prevalence of advanced fibrosis (stages 3 and 4) in T2D patients is 17% by liver biopsy, 7.3–25.0% by FibroScan and 4.3–7.1% by magnetic resonance elastography (MRE) [9]. NAFLD is closely associated with diabetic nephropathy, chronic kidney disease (CKD) and end-stage renal disease (ESRD). Because NASH-associated cirrhosis often complicates CKD or ESRD, simultaneous kidney liver transplantation (SKLT) is dramatically increasing in the US [10]. According to the National Health and Nutrition Examination Survey (NHANES) 1999–2016, NAFLD with renal insufficiency shows significantly higher mortality than only NAFLD [11]. NASH is the most rapidly growing indication for SKLT [12]. Recipients with NASH or cryptogenic cirrhosis with a body mass index (BMI) greater than 30 showed a lower estimated glomerular filtration rate (eGFR) and higher graft loss after SKLT compared with those with other chronic liver disease [10]. The belief that NAFLD and CKD share common pathogenetic mechanisms for progression leads to the hypothesis that they can also share potential therapeutic targets. We here review common drug pipelines for diabetic nephropathy and hepatopathy.

## 2. The Association of NASH/NAFLD with Diabetic Nephropathy/CKD

NAFLD and CKD are global public health problems, affecting up to 25–30% and up to 10–15% of the general population for NAFLD and CKD, respectively. Diabetic nephropathy is the leading cause of CKD and ESRD. Recently, it has also been established that there is a strong association between NAFLD and CKD, regardless of the presence of potential confounding diseases such as obesity, hypertension and T2D. Since NAFLD and CKD are both common diseases that often occur alongside other metabolic conditions, such as T2D or metabolic syndrome, elucidating the relative impact of NAFLD on the risk of incident CKD presents a substantial challenge for investigators working in this research field. A growing body of epidemiological evidence suggests that NAFLD is an independent risk factor for CKD, and recent evidence also suggests that associated factors such as metabolic syndrome, dysbiosis, unhealthy diets, platelet activation and processes associated with aging could also contribute mechanisms linking NAFLD and CKD [13] (Figure 1). Accumulating evidence has proved that NAFLD is associated with diabetic nephropathy, independent of confounding factors [14]. Liver fibrosis—but not steatosis—was found to be independently associated with albuminuria in 1763 Chinese patients with T2D [15]. We previously reported that patients with biopsy-proven NASH were more likely to have CKD than patients without NASH [16]. The presence and severity of NAFLD has been related to the incidence and stage of CKD [17] independent of traditional CKD risk factors; conversely, the presence of CKD increases overall mortality in NAFLD patients [18]. In accordance with the pathogenic link between NAFLD and CKD, NASH-related cirrhosis carries a higher risk of ESRD than other etiologies of cirrhosis; furthermore, it is an increasing indication for SKLT and an independent risk factor for kidney graft loss and cardiovascular disease (CVD) [12,19]. NAFLD was accompanied by a higher risk of incident CKD (hazard ratio (HR): 1.22, 95% confidence interval (CI) 1.04–1.43) in a retrospective cohort study of 41,430 adult men and women without CKD at baseline. The risk of CKD increased progressively with increased NAFLD severity, which was evaluated by the NAFLD fibrosis score (NFS) [20]. In a Japanese retrospective study, the fibrosis-4 (FIB-4) index and the presence of T2D were significant risk factors for CKD development [21]. In that study, the patatin-like phospholipase domain-containing protein-3 (PNPLA3) genotype was not related to CKD. In contrast, a recent study from Italy found a relationship between the PNPLA3 genotype and incident CKD [22]. A meta-analysis showed that NAFLD was associated with a nearly 40% increase in the long-term risk of incident CKD [23]. Based on a cohort study of 1525 CKD patients, the annual change in decline in eGFRs in CKD patients with NAFLD was larger than those without NAFLD. The decline in eGFRs associated with NAFLD was greater in patients with a higher NFS, in those with proteinuria or with a low eGFR at baseline (<45 mL/min/1.73 m^2^) and in those who were smokers and hypertensive [24]. Collectively, these data suggest that common pathogenic mechanisms underlie both liver and kidney injury and could be targeted to retard the progression of both NAFLD and CKD (Figure 1).

An accumulation of visceral fat, which is closely associated with NAFLD and CKD, causes chronic inflammation. Visceral adipose tissue increases in plasma concentrations of nonesterified fatty acids (NEFAs). With the increase in the supply of NEFA to the liver, hepatic macrophages are activated. The activation of hepatic macrophages and hepatic inflammation is associated with an increase in proinflammatory cytokines and hepatic/systemic insulin resistance, increased activity of the renin–angiotensin–aldosterone system (RAAS) and oxidative stress mediated by proinflammatory and profibrotic mediators. In turn, the kidney reacts, promoting further RAAS activation, increased angiotensin II and uric acid production in a vicious cycle leading to hepatic fibrosis progression. Excessive dietary fructose intake also affects renal injury through altered lipogenesis and inflammatory response. Experimental evidence also supports a role of the inflammasome and innate immune system in NAFLD and CKD [25,26,27,28]. In this way, NAFLD and CKD share common proinflammatory and profibrotic mechanisms of disease progression (Figure 1). Therefore, all of these pathways indicate a causal link between NAFLD and CKD, whereby NAFLD increases the risk of incident CKD.

Although there are no established pharmacotherapies for advanced stages of NAFLD and CKD, a variety of drug pipelines for liver and renal injury exist [29]. The modulation of nuclear transcription factors regulating key pathways of lipid metabolism, including peroxisome proliferator-activated receptors (PPARs) and farnesoid X receptor (FXR), is now under stage 3 clinical development [30]. Other therapeutic approaches target key mediators of inflammation, fibrogenesis, gut dysfunction through gut microbiota manipulation and antidiabetic therapies. Furthermore, NAFLD affects CKD per se through lipoprotein metabolism and hepatokine secretion, and conversely, targeting the renal tubule by sodium–glucose cotransporter 2 (SGLT2) inhibitors can improve both CKD and NAFLD.

## 3. Common Drug Pipelines for NAFLD/NASH and Diabetic Nephropathy/CKD

### 3.1. Metabolic Modifiers

#### 3.1.1. Peroxisome Proliferator-Activated Receptors (PPARs)

PPARs are nuclear receptors that are involved in the transcriptional regulation of lipid metabolism, energy balance, insulin metabolism, inflammation and atherosclerosis. Three isotypes of PPARs exist: PPAR-α, PPAR-δ and PPAR-γ [31]. PPAR-α is expressed ubiquitously, but is most highly expressed in the liver. It plays a critical role in the regulation of fatty acid uptake, beta oxidation, ketogenesis, bile acid synthesis and triglyceride turnover. PPAR-α is also thought to have anti-inflammatory effects through the complex regulation of nuclear factor kappa B (NF-κB). PPAR-δ is expressed in skeletal muscle, adipose tissue and skin, but it is most highly expressed in muscle, where it is involved in regulating mitochondrial metabolism and fatty acid beta oxidation [32]. PPAR-δ is well expressed in hepatocytes but is also expressed in Kupffer cells and hepatic stellate cells (HSCs), suggesting a potential role in inflammation and fibrosis [32]. PPAR-γ is most highly expressed in adipose tissue, where it serves an essential role in the regulation of adipocyte differentiation, adipogenesis and lipid metabolism. PPAR-γ activation results in the increased production of various adipokines, including adiponectin, which enhances hepatic fatty acid oxidation. In addition to its metabolic effects, PPAR-γ agonists are also thought to decrease inflammation and cytokine production in patients with metabolic syndrome [32].

Pioglitazone (a PPAR-γ agonist) showed a proven histological improvement in NASH compared to a placebo [33,34,35,36]. The PPAR-γ agonist also has a protective effect against various types of injury of the kidney including diabetic and nondiabetic kidney disease [37]. However, this agent has several safety concerns, including edema, heart failure, cancer incidence and osteoporosis in women. The India-based Zydus Cadila is evaluating the once-daily oral experimental therapy saroglitazar magnesium for NASH patients in a phase 2 trial. Saroglitazar is a dual PPARα/γ agonist that is approved in India for the treatment of dyslipidemia in diabetic patients [38]. In real-world clinical studies with a duration of up to 58 weeks, saroglitazar effectively improved lipid and glycemic parameters without significant adverse effects (AEs) in 5824 patients with diabetic dyslipidemia [39]. In mice with choline-deficient high-fat-diet-induced NASH, saroglitazar reduced hepatic steatosis, inflammation and ballooning and prevented fibrosis development. It also reduced serum alanine aminotransferase (ALT), aspartate aminotransferase (AST) and inflammatory and fibrosis biomarker expressions [40]. In this model, the reduction in the overall NAFLD activity score (NAS) due to saroglitazar (3 mg/kg) treatment was significantly more prominent than that due to pioglitazone (25 mg/kg) or fenofibrate (100 mg/kg) [41]. A phase 2, randomized double-blind placebo-controlled trial (RDBPCT) comparing three doses of saroglitazar (1, 2 and 4 mg) with a placebo in NAFLD is ongoing (EVIDENCES IV, Table 1). This study enrolled 104 patients with NAFLD/NASH. The primary endpoint is the percentage change from baseline in serum ALT levels at week 16 in the saroglitazar groups compared with the placebo group. At the Liver Meeting 2019, Gawrieh and colleagues showed that patients in the saroglitazar groups (*n* = 77) exhibited significantly reduced ALT levels compared to those in the placebo group (*n* = 27). The absolute change in liver fat content by MRI-proton density fat fraction (MRI-PDFF) from the baseline to week 16 was significantly greater (−4.21%) in the saroglitazar 4 mg group than in the other groups [42]. Aleglitazar, a dual PPARα/γ agonist, slowed eGFR decline in stage 3 diabetic CKD (phase 2b, AleNephro) [43]. Patients were randomized for a 52 week double-blind treatment with aleglitazar at 150 μg/d (*n* = 150) or pioglitazone at 45 mg/d (*n* = 152). The mean eGFR change from baseline to the end of follow-up was −2.7% (95% CI: −7.7, 2.4) with aleglitazar versus −3.4% (95% CI: −8.5, 1.7) with pioglitazone, establishing noninferiority (0.77%; 95%CI: −4.5, 6.0) [43].

Elafibranor, a PPAR α/δ dual agonist, inhibits CKD progression in NASH mice [44]. A multicenter phase 3 RDBPCT is ongoing to evaluate the efficacy and safety of elafibranor (120 mg/d) in NASH patients with stage 2/3 fibrosis and NAS ≥ 4 (RESOLVE-IT, Table 1). The primary outcomes of this study are to evaluate the effect of elafibranor treatment compared with placebo on (1) histological improvement (resolution of NASH without worsening of fibrosis at 72 weeks) and (2) composite long-term outcomes, composed of all-cause mortality, cirrhosis and liver-related clinical outcomes. After 72 weeks of treatment, the study missed its primary endpoint, with 19% of patients in the treatment arm achieving NASH resolution without fibrosis getting worse, compared to 15% of patients in the placebo group. Only 25% of elafibranor patients showed fibrosis improvement by at least one stage, compared to 22% of placebo patients. A phase 1 study is being conducted in order to assess the need for dose adjustment for elafibranor in participants with renal impairment. Pharmacokinetic parameters of elafibranor and its active metabolite (GFT1007) will be compared in severe renally impaired participants (eGFR < 15 mL/min/1.73 m^2^) versus healthy participants after a single oral administration of elafibranor at 120 mg (Table 1).

#### 3.1.2. Farnesoid X Receptor Agonist

Obeticholic acid (OCA), a semisynthetic analog of chenodeoxycholic acid, is an FXR agonist. FXR is a nuclear receptor that is highly expressed in the liver and small intestine. Bile acids are natural ligands of FXR, and their binding with and activation of FXR is critical to the regulation of cellular pathways that modulate BA synthesis, lipid metabolism, inflammation and fibrosis. OCA markedly suppresses hepatocyte death and liver fibrosis with only marginal effects on body weight and hepatic steatosis in a murine model of NASH [45]. An international phase 3 study (REGENERATE study) for NASH patients is ongoing. Interim analyses showed that OCA at 25 mg/d for 72 weeks significantly ameliorated hepatic fibrosis (≥1 stage fibrosis) compared with a placebo [46]. A phase 3 trial of OCA for cirrhotic patients due to NASH is ongoing (REVERSE trial) [30]. This trial enrolled 540 NASH cirrhotic patients and is being conducted at sites in North America, Europe, Australia and New Zealand. The primary endpoint is the percentage of patients with fibrosis improvement (more than one stage) after one year of treatment. Patients were randomized in a 1:1:1 ratio to one of the three treatment arms: OCA 10 mg/d, OCA 10 mg/d with titration to 25 mg/d at three months, or placebo. Patients who complete the double-blind phase of the REVERSE trial will be eligible to enroll in an open-label extension study for up to 12 additional months (Table 1).

OCA has been shown to reverse renal lipid accumulation, proteinuria and tubulo-interstitial inflammation and fibrosis in diet-induced experimental CKD [47,48,49,50,51]. Clinical studies of OCA for diabetic nephropathy/CKD have not yet been planned.

### 3.2. Antioxidative Agents

Oxidative stress is considered to be a key mechanism of hepatocellular injury and disease progression in patients with NASH [52] and CKD [53]. The transcription factor Nrf2 (nuclear factor-erythroid-2-related factor 2) plays a central role in stimulating the expression of various antioxidant-associated genes in the cellular defense against oxidative stress [54].

#### 3.2.1. Oltipraz

Oltipraz, 5-(2-pyrazynyl)-4-methyl-1,2-dithiole-3-thione, is a synthetic dithiolethione that targets Nrf2, an agent that plays a pivotal role in the cellular defense against oxidative stress by promoting the transcription of various antioxidant genes [55]. A phase 2a study showed that 24-week oltipraz treatment significantly reduced the liver fat content in patients with NAFLD (PMK-N01GI1) [56] (Table 1). Oltipraz ameliorated renal fibrosis in a unilateral ureteral obstruction rat model [57]. However, human studies for diabetic nephropathy or CKD are not planned.

#### 3.2.2. Bardoxolone Methyl

Bardoxolone methyl is a semisynthetic triterpenoid that is derived from the natural product oleanolic acid and is known to be one of the most potent inducers of Nrf2 [58,59,60,61]. Bardoxolone methyl was associated with an improvement in the eGFR in patients with advanced CKD (defined as an eGFR of 20–45 mL per minute per 1.73 m^2^ of body surface area) and T2D at 24 weeks (BEAM trial) [62]. Among patients with T2D and stage 4 CKD, however, bardoxolone methyl did not reduce the risk of ESRD or death from cardiovascular causes. A higher rate of cardiovascular events with bardoxolone methyl than with placebo prompted the termination of a trial (BEACON study) [63,64]; however, a multicenter phase 2 RDBPCT in Japan enrolled 124 patients with CKD (stage G3 and G4) and T2D without identified risk factors for fluid overload, such as a baseline brain natriuretic peptide (BNP) count >200 pg/mL and prior history of heart failure (TSUBAKI study) [65]. The interim analysis of this trial demonstrated a significant improvement in the eGFR in the bardoxolone methyl group compared with the eGFR in the placebo group without safety concerns. A phase 3 study of bardoxolone methyl in patients with DKD (stage G3 and G4) is ongoing (AYAME study, Table 1). This trial will enroll 700 patients. Bardoxolone methyl prevented the development of insulin resistance and hepatic steatosis in mice fed a high-fat diet [66]; however, the use of this agent is not planned for NASH/NAFLD.

### 3.3. Anti-Inflammatory and Antiapoptosis

#### 3.3.1. C-C Motif Chemokine Receptor-2/5 Inhibitor

Cenicriviroc is an oral inhibitor of C-C chemokine receptor types 2 (CCR2) and 5 (CCR5) which plays an important role in the hepatic recruitment of macrophages [67,68]. Macrophage recruitment through CCR2 into adipose tissue is believed to play a role in the development of insulin resistance and T2DM. The administration of CCR2 antagonist modestly improved glycemic parameters compared with a placebo [69]. CCR5 antagonist is expected to impair the migration, activation and proliferation of collagen-producing HSCs [70]. In animal models, cenicriviroc showed antifibrotic effects, with significant reductions in collagen deposition (*p* < 0.05) and collagen type 1 protein and mRNA expression in liver and kidney [71].

According to a phase 2b trial (CENTAUR study), fibrosis improved significantly without NASH worsening after one year of cenicriviroc treatment (20%) compared with a placebo (10%) [72]. Although asymptomatic amylase elevation (grade 3) was more frequent in the cenicriviroc group than in the placebo group, this agent was well-tolerated. No significant improvement of fibrosis without worsening NASH after two years of cenicriviroc treatment was found (35%) compared with a placebo (20%) [73]. A phase 3 study is ongoing to evaluate the effects of cenicriviroc on hepatic fibrosis in 2000 patients with NASH (AURORA study) [74] (Table 1)**.** A phase 2a, multicenter RDBPCT of cenicriviroc is being conducted with approximately 50 adult obese subjects (BMI ≥ 30 kg/m^2^) with prediabetes or T2D and suspected NAFLD (ORION study).

The small-molecule CCR2 antagonist CCX140-B was shown to reduce albuminuria and slow eGFR decline in diabetic nephropathy [75]. The dual chemokine receptor CCR2/CCR5 antagonists (BMS-813160 and PF-04634817) were evaluated in diabetic nephropathy. However, clinical development for this indication was discontinued in light of the modest efficacy observed, although PF-04634817 appeared to be safe and well-tolerated [76].

#### 3.3.2. Apoptosis Signaling Kinase-1 Inhibitor

Apoptosis signal-regulating kinase 1 (ASK1) is activated by extracellular tumor necrosis factor alpha (TNFα), intracellular oxidative or ER stress and initiates the p38/JNK pathway, resulting in apoptosis and fibrosis [77]. The inhibition of ASK1 has, therefore, been proposed as a target for the treatment of NASH [78]. Thus, international phase 3 trials evaluating a selective ASK1 inhibitor (selonsertib) among NASH patients with stage 3 (STELLAR3) or cirrhosis (STELLAR4) were initiated (Table 1). Unfortunately, the STELLAR trial was discontinued because selonsertib did not meet the primary endpoint [79]. STELLAR4 found that 14.4% of patients treated with selonsertib at 18 mg (*p* = 0.56 versus placebo) and 12.5% treated at the lower 6 mg dose (*p* = 1.00) achieved at least a ≥1-stage improvement in fibrosis, compared with 12.8% of placebo recipients. In the STELLAR3 trial of 802 enrolled patients, 9.3% of patients treated with selonsertib 18 mg (*p* = 0.42 vs. placebo) and 12.1% of patients treated with selonsertib 6 mg (*p* = 0.93) achieved a ≥1-stage improvement in fibrosis without worsening of NASH after 48 weeks of treatment, versus 13.2% with a placebo.

ASK1 activation in glomerular and tubular cells resulting from oxidative stress may drive kidney disease progression [80]. Findings in animal models identified selonsertib as a potential therapeutic agent [81]. The primary objective of a phase 2 study was to determine the effect of selonsertib on eGFR decline in 334 participants with T2D and treatment-refractory moderate-to-advanced DKD. Participants were randomized with a 1:1:1:1 allocation to receive one of three doses of selonsertib (2 mg, 6 mg, or 18 mg) or a matching placebo. The primary outcome was the change from baseline eGFR at 48 weeks [82]. Although the trial did not meet its primary endpoint, post hoc analyses found that between 4 and 48 weeks, the rate of eGFR decline was reduced 71% for the 18 mg group relative to a placebo. A phase 3, RDBPCT evaluating the efficacy and safety of selonsertib in subjects with moderate-to-advanced DKD is ongoing (MOSAIC study, Table 1). The primary objective of this study is to evaluate whether selonsertib can slow the decline in kidney function, reduce the risk of kidney failure or reduce the risk of death due to kidney disease in 3300 participants with DKD.

### 3.4. Antifibrotic Agent

Galectin-3 (Gal-3) is a β-galactoside-binding lectin secreted in the disease state, mainly secreted by macrophages [83]. It binds to the cell surface and extracellular matrix glycans and affects a variety of physiologic processes, including cell apoptosis, adhesion, migration, angiogenesis and inflammatory responses [84]. Gal-3 protein expression, which is required for the development of hepatic fibrosis, was increased in NASH, with the highest expression in macrophages surrounding lipid-laden hepatocytes [85]. Elevated plasma levels of Gal-3 were also associated with increased risks of rapid renal function decline, incident CKD and progressive renal impairment, as well as with CVD events, infection and all-cause mortality in patients with renal function impairment [86].

#### 3.4.1. Belapectin

Belapectin (GR-MD-02, Galectin Therapeutics Inc. Norcross, GA, USA), a Gal-3 antagonist, markedly improved liver histology with significant reductions in NAS and fibrosis in mice models [87]. Although there were no safety concerns in a phase 2a trial of NASH patients with stage 3 fibrosis [88], there was no apparent improvement in the three noninvasive tests for the assessment of liver fibrosis. A phase 2b clinical trial to evaluate the safety and efficacy of belapectin for the treatment of liver fibrosis and resultant portal hypertension in 162 patients with NASH cirrhosis (NASH-CX trial) was completed [89] (Table 1). In the phase 2b trial, dubbed NASH-CX, belapectin was administered as an infusion every other week for 52 weeks, for a total of 26 doses. Approximately half of the NASH cirrhosis patients in the trial had esophageal varices, and the other half of the subjects were without esophageal varices. The NASH-CX trial missed the primary endpoint of reaching statistical significance in reducing the hepatic venous pressure gradient (HVPG), when the total group of patients was considered. However, a statistically significant and clinically meaningful effect of belapectin was observed for the primary endpoint measurement of HVPG in the subgroup of NASH cirrhosis patients without esophageal varices. The company plans to advance belapectin to phase 3 testing for NASH cirrhosis patients without esophageal varices.

#### 3.4.2. GCS-100

On the other hand, serum levels of Gal-3 levels were associated with an increased risk of all-cause mortality and cardiovascular events in patients with CKD [90]. In patients with T2D, the mean levels of Gal-3 were significantly higher in patients with macroalbuminuria (urinary albumin/creatinine ratio (ACR) = >300 mg/g) than in those with microalbuminuria (30–300 mg/g) and normoalbuminuria (ACR = <30 mg/g) [91]. Gal-3 inhibition attenuates renal injury progression in cisplatin-induced nephrotoxicity [92]. Thus, Gal-3 antagonist will become a therapeutic option for diabetic nephropathy/CKD. A phase 2b RDPBCT of GCS-100 in patients with CKD caused by diabetes will enroll approximately 375 patients at multiple centers located in the US (Table 1); the study duration is six months. Patients will be randomly assigned 1:1:1:1 to a treatment with placebo or 1 mg, 3 mg or 9 mg GCS-100. All doses of the study drug will be administered via intravenous push injection once weekly for two months (eight weeks), then every other week for an additional four months (16 weeks).

### 3.5. Antihypertensive Agents

#### 3.5.1. Angiotensin-Converting Enzyme Inhibitors and Angiotensin Receptor Blockers

The use of these medications in CKD has been extensively evaluated, and based on the collaborative study group trial and several others, the use of angiotensin-converting enzyme inhibitors (ACE-I) and angiotensin receptor blockers (ARBs) in patients with CKD with proteinuria is now a level-one recommendation by the Kidney Disease Outcomes Quality Initiative (KDOQI) [93]. ARBs, a class of antihypertensive drugs, are potential therapeutic agents for NAFLD because of their anti-inflammatory or antifibrotic actions [94]. Telmisartan, which is an ARB with PPAR-regulating activity, was compared to the use of valsartan in the fatty liver protection by telmisartan (FANTASY) trial and found to cause a reduction in necroinflammation, NAFLD activity score (NAS) and fibrosis stage in NASH, as well as microalbuminuria [95]. However, current evidence is insufficient to support the efficacy of ARBs in managing fibrosis in NAFLD patients [96].

#### 3.5.2. Nonsteroidal Mineral Corticoid Receptor Antagonist

Aldosterone is a mineralocorticoid hormone with a well-known effect on the renal tubule leading to water retention and potassium reabsorption [97]. Other major effects of the hormone include the induction of proinflammatory activity, which leads to the progressive fibrotic damage of the target organs, heart and kidney. Blocking the aldosterone receptor, therefore, represents an important pharmacological strategy to avoid the clinical conditions arising from NASH [98,99] and CKD [100,101,102]. Apararenone (MT-3995) [103] is a nonsteroidal antimineralocorticoid which is under development for the treatment of diabetic nephropathies and NASH (Table 1). An exploratory phase study of aparerenone in 48 Japanese patients with biopsy-proven NASH (which was a placebo-controlled double-blind study) was completed [103]. The primary endpoint was the percentage change from baseline in ALT. A phase 2 RDBPCT of aparerenone (low dose) in subjects with diabetic nephropathy was completed [103] (Table 1). Another phase 2 RDBPCT evaluated the effect on ACR, the pharmacodynamics, safety, tolerability and pharmacokinetics of multiple oral doses of aparerenone as an add-on therapy to ACE-I or ARB in T2D nephropathy subjects with albuminuria and an eGFR of ≥30 to <60 mL/min/1.73 m^2^. A long-term study of aparerenone has also been completed to evaluate drug safety; however, these results have never been published.

### 3.6. Anti-Diabetic Agents

#### 3.6.1. Glucagon-Like Peptide Receptor Agonist

Glucagon-like peptide (GLP-1) is a gut-derived incretin hormone that induces insulin secretion and reduces glucagon secretion in a glucose-dependent manner, suppresses appetite and delays gastric emptying [104,105]. GLP-1 receptor agonists (GLP-1 RAs) are expected to be an attractive therapeutic option for T2D patients with NASH. GLP-1 RAs have been shown to reduce liver enzymes and oxidative stress and improve liver histology in murine NASH models [106,107].

A phase 2 study showed that liraglutide showed histological improvement in NASH patients (LEAN study). The mechanisms of a GLP-1 RA for NASH can be explained not only by weight loss and diabetic control but also by potent anti-inflammatory activity [108]. A phase 3 open-label study is ongoing to compare the effects of liraglutide and bariatric surgery on weight loss, liver function, body composition, insulin resistance, endothelial function and biomarkers of NASH in obese Asian adults (CGH-LiNASH). Dulaglutide has some advantages, such as weekly injection, disposable and prefilled devices and safety profiles similar to those of other GLP-1 RAs. The D-LIFT (effect of dulaglutide on liver fat) trial is a prospective open-label randomized controlled trial (RCT) to examine the effect of dulaglutide 0.75 mg subcutaneously weekly for four weeks, followed by 1.5 mg weekly for 20 weeks when included in the standard treatment for T2D vs. standard treatment for T2D (minus dulaglutide) in T2D patients with NAFLD. Hepatic steatosis will be measured by magnetic resonance imaging-proton density fat fraction (MRI-PDFF). Semaglutide, a novel GLP-1 RA, is now approved for diabetic patients in the US, EU, Canada and Japan. A phase 2 RDBPCT to compare the efficacy and safety of three different doses of semaglutide (once-daily subcutaneous injection) versus placebo in 288 participants with NASH (stage 1–3 fibrosis) is ongoing (SEMA-NASH study, Table 1).

GLP-1RAs may exert beneficial actions on the kidneys by lowering glucose and blood pressure, decreasing insulin levels and causing weight loss. Emerging evidence suggests potential protective actions of GLP-1RAs on the kidneys, independently of their glucose-lowering effects, some of which may play a role in the inhibition of development and progression of DKD [109,110]. In humans, GLP-1R has been identified in the kidney, localized in proximal tubular cells and preglomerular vascular smooth muscle cells [111]. However, the precise mechanisms of the renoprotective effects of GLP-1RA remain unknown. A recent meta-analysis of seven trials consisting of ELIXA (lixisenatide) [112], LEADER (liraglutide) [113], SUSTAIN-6 (semaglutide) [114], EXSCEL (exenatide) [115], Harmony outcomes (albiglutide) [116], REWIND (dulaglutide) [117] and PIONEER 6 (oral semaglutide) [118] showed that GLP-1RA treatment has beneficial effects on cardiovascular, mortality and kidney outcomes in T2DM [119]. According to a post hoc analysis that evaluated the safety of liraglutide treatment in patients with CKD in LEADER, the use of liraglutide in patients with CKD was safe, with no difference between patients with and without CKD [119]. As a result, GLP-1 RAs are most promising for the treatment of NASH with CKD.

#### 3.6.2. Sodium–Glucose Cotransporter Inhibitor 2

SGLT2 inhibits glucose reabsorption in the proximal tubule, leading to glucouria and plasma glucose reduction. Therefore, SGLT2 inhibitors have become promising therapeutic agents in NASH and NAFLD patients [120]. Several pilot studies or open randomized controlled trials (RCTs) have found a significant reduction in transaminase activity, body weight, hepatic steatosis, fatty liver index and liver histology (steatosis and fibrosis) in NAFLD patients [121,122,123,124,125,126,127,128]. Not only HbA1c and transaminase activities but also the hepatic fat content evaluated by MRIhepatic fat fraction were significantly decreased after 24 weeks of therapy with luseogliflozin (LEAD trial) [129]. In the E-LIFT trial, 50 T2D patients with NAFLD (≥40 years old) were randomized to empagliflozin (10 mg/d) plus their standard medical treatment for T2D, such as metformin and/or insulin, or to the receipt of only their standard treatment without empagliflozin (control group). After 20 weeks of treatment, the liver fat content measured by using MRI-PDFF of the group receiving empagliflozin decreased from an average of 16.2% to 11.3% (*p* < 0.0001), whereas the control group had only a decrease from 16.4% to 15.6% (*p* = 0.057) [130]. To evaluate the histological efficacy and safety of dapagliflozin in NASH, a phase 3 RDBRCT (dapagliflozin efficacy and action in NASH (DEAN) study) is now recruiting and will enroll 100 participants (Table 1). In the CREDENCE trial [131], patients with T2D and albuminuric CKD were randomly assigned to receive canagliflozin, an oral SGLT2 inhibitor, at a dose of 100 mg daily, or a placebo. All the patients had an eGFR of 30 to <90 mL per minute per 1.73 m^2^ of body surface area and albuminuria (ACR, >300 to 5000 mg/g) and were treated with a renin–angiotensin system blockade. The primary outcome was a composite of ESRD (dialysis, transplantation, or a sustained eGFR of <15 mL per minute per 1.73 m^2^), a doubling of the serum creatinine level or death from renal or cardiovascular causes. The relative risk of the primary outcome was 30% lower in the canagliflozin group than in the placebo group, with event rates of 43.2 and 61.2 per 1000 patient-years, respectively (hazard ratio (HR), 0.70; 95% CI, 0.59 to 0.82; *p*  =  0.00001) [95]. Thus, SGLT2 inhibitor showed renoprotective efficacy. Recent Western guidelines recommended the use of SGLT2 inhibitor in T2D patients with CKD (eGFR 30 to ≤60 mL min^−1^ [1.73 m]^−2^ or ACR >30 mg/g, particularly >300 mg/g) [132]. Two trials (DAPA-CKD [133], EMPA-KIDNEY are ongoing to explore the renoprotective efficacy of SGLT2 inhibitors for CKD patients without T2D (Table 1).

### 3.7. Gut Microbiome (Gut–Liver–Kidney Axis)

The ability of the gut to modulate the host metabolism and inflammatory response and its contribution to obesity-related complications, including NAFLD and CKD, has been increasingly recognized [134,135], and various gut-oriented approaches to treat NASH and CKD are under evaluation, including the modulation of gut microbiota and of gut-derived peptide incretins and fibroblast growth factor 19. Two main strategies are being evaluated to counteract host adverse effects of dysregulated gut microflora: the first involves the modulation of gut microbiota composition, and the second is the direct antagonization of microbial proinflammatory mediators. In a randomized trial of 104 patients with NAFLD, one year of administration of a synbiotic combination (probiotic and prebiotic) altered fecal microbiomes but did not reduce liver fat content or markers of liver fibrosis (INSYTE) [136,137] (Table 1). A meta-analysis evaluating 28 clinical trials showed that probiotics are superior to placebos in NAFLD patients and could be utilized as a common complementary therapeutic approach [138]. A pilot study suggested probiotic dietary supplements are more effective than a placebo in reducing blood urea nitrogen (BUN) and improving the quality of life of patients with stage 3 or 4 CKD [139]. The impact of gut microbiota manipulation with probiotics, prebiotics or synbiotics on renal function in CKD was investigated in the synbiotics easing renal failure by improving gut microbiology (SYNERGY) study [140] (Table 1). This trial found that synbiotics decreased serum p-cresyl sulfate without reducing serum indoxyl sulfate in nondialysis CKD. Another systematic review found that prebiotic and probiotic therapies reduced indoxyl sulfate and p-cresyl sulfate in patients with ESRD on hemodialysis [141]. However, it is unclear whether the results hold true for other patients with CKD.

## 4. Conclusions

NASH/NAFLD is closely associated with diabetic nephropathy/CKD. To prevent morbidity and mortality in T2D patients, they should be considered for pharmacotherapies in addition to conventional dietary interventions. Because SLKT, which is now increasing in the US, will result in unacceptably high morbidity and public healthcare costs, a variety of drug pipelines exist for simultaneously treating NASH/NAFLD and diabetic nephropathy/CKD, such as PPAR agonists, FXR agonists, CCR2/5 antagonists, Nrf2 activators, ASK-1 inhibitors, Gal-3 inhibitors and gut microbiome manipulation. Unfortunately, several clinical studies have been discontinued due to insufficient evidence or adverse effects. Since NASH/NAFLD is considered to be a multifactorial disease, the importance of combining therapies that engage with different targets and which have synergistic benefits for individual therapies has been highlighted. NASH/NAFLD patients with T2D should be preferentially treated with novel drugs licensed for diabetes treatment such as GLP-1RA and SGLT2 inhibitors [120], because these agents also have hepatoprotective [146] and nephroprotective efficacy [119,131]. Among a variety of SGLT2 inhibitors, dapagliflozin has entered phase 3 trials for patients with biopsy-proven NASH (DEAN study) or nondiabetic CKD (DAPA-CKD study) [133]. Cost-effectiveness data and patient-reported outcome benefits are also required for companies to position their medications within practical NASH or CKD guidelines. It is expected that one approach will solve both problems (diabetic hepatopathy and nephropathy) in the near future.

## Figures and Tables

**Figure 1 ijms-21-04939-f001:**
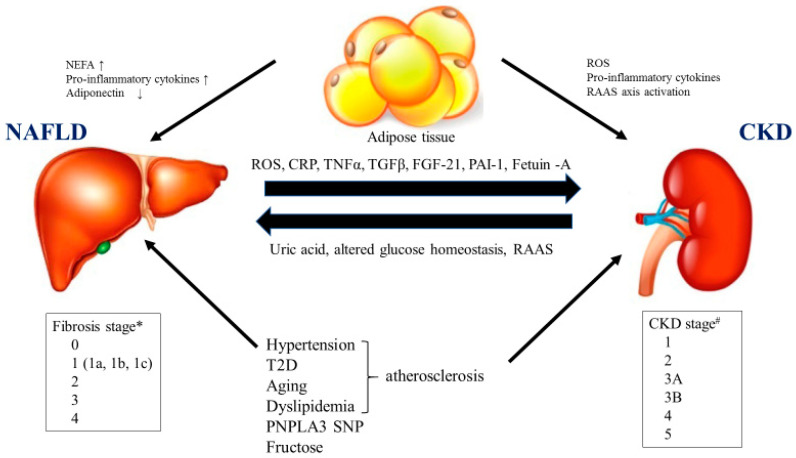
Potential mechanisms linking nonalcoholic steatohepatitis (NASH)/nonalcoholic fatty liver disease (NAFLD) and chronic kidney disease (CKD)/diabetic nephropathy. NEFA: nonesterified fatty acid, ROS: reactive oxygen species, RAAS: renin–angiotensin–aldosterone, CRP: C-reactive protein, TNFα: tumor necrosis factor α, TGFβ: transforming growth factorβ, FGF-21: fibroblast growth factor-21, PAI-1: plasminogen activator inhibitor-1, T2D: type 2 diabetes, PNPLA3: patatin-like phospholipase domain-containing protein-3, SNP: single nucleoside polymorphism. * NASH Clinical Research Network. # National Kidney Foundation.

**Table 1 ijms-21-04939-t001:** Common drug pipelines for NASH/NAFLD and CKD/diabetic nephropathy.

Action	Drug	NASH/NAFLD	CKD/Diabetic Nephropathy
**(1) Metabolic modifiers**			
PPARα/γ agonist	Saroglitazar	▪ Phase 2 * EVIDENCES IV [42] (NCT03061721)	―
	Aleglitazar	―	▪ Phase 2b * AleNephro [43] (NCT01043029)
PPARα/δ agonist	Elafibranor	▪ Phase 3 # RESOLVE-IT (NCT02704403)	▪ Phase 1 # (NCT03844555)
FXR agonist	Obeticholic acid	▪ Phase 3 # REGENERATE [46] (NCT02548351) ▪ Phase 3 # REVERSE (NCT03439254)	▪ Preclinical [50,51]
**(2) Antioxidants**			
Nrf2 activator	Oltipraz	▪ Phase 2a * PMK-N01GI1 [56] (NCT01373554)	―
	Bardoxolone methyl	―	▪ Phase 2 * BEAM [62] (NCT00811889) ▪ Phase 2 $ BEACON [63] (NCT01351675) ▪ Phase 2a # TSUBAKI [65] (NCT02316821) ▪ Phase 3 # AYAME (NCT03550443)
**(3) Anti-inflammatory and antiapoptosis**			
CCR2/5 antagonist	Cenicriviroc	▪ Phase 2b * CENTAUR [72,73] (NCT02217475) ▪ Phase 2a # ORION (NCT02330549) ▪ Phase 3 # AURORA [74] (NCT03028740)	―
	BMS-813160	―	▪ Phase 2a $ (NCT01752985)
	PF-04634817	―	▪ Phase 2 $ (NCT01712061)
ASK1 inhibitor	Selonsertib	▪ Phase 3 $ STELLAR 3/4 [79] (NCT03053050) (NCT03053063)	▪ Phase 2 * [82] (NCT02177786) ▪ Phase 3 # MOSAIC (NCT04026165)
**(4) Antifibrotic agent**			
Galectin-3 antagonist	Belapectin	▪ Phase 2b * NASH-CX [89] (NCT02462967)	―
	GCS-100	―	▪ Phase 2b # (NCT02312050)
**(5) Antihypertensive drugs**			
Nonsteroidal MRAs	Aparerenone (MT-3995)	▪ Phase 2 * (NCT02923154)	▪ Phase 2 * (NCT02205372) (NCT01756716) (NCT02676401)
**(6) Antidiabetic agents**			
GLP-1RA	Liraglutide	▪ Phase 2 * LEAN [107] (NCT02654665) ▪ Phase 3 # CGH-LiNASH (NCT02654665)	▪ LEADER trial * [113] (NCT01179048)
	Exenatide	―	▪ Phase 2a * [142,143]
	Dulaglutide	▪ D-LIFT # (NCT 03590626)	▪ REWIND * [117] (NCT01394952) ▪ AWARD7 * [144] (NCT 01621178)
	Semaglutide	▪ Phase 2 # SEMA-NASH (NCT02970942)	―
SGLT2 inhibitor	Dapagliflozin	▪ Phase 3 # DEAN (NCT03723252)	▪ DAPA-CKD * [133] (NCT03036150)
	Canagliflozin	―	▪ CREDENCE * [131] (NCT02065791)
	Empagliflozin	▪ E-LIFT [130] * (NCT02686476)	▪ A Slope Analysis from the EMPA-REG OUTCOME * [145] ▪ EMPA-KIDNEY # (NCT03594119)
**(7) Gut microbiota manipulation**			
Prebiotics Probiotics Synbiotics		▪ INSYTE [136,137] * (NCT01680640)	▪ Phase 2a * SYNERGY [140] (ACTRN1261300049)

NASH: nonalcoholic steatohepatitis, NAFLD: nonalcoholic fatty liver disease, SGLT2: sodium–glucose cotransporter 2, GLP-1RA: glucagon-like peptide receptor agonist, PPAR: peroxisome proliferator-activated receptor, FXR: farnesoid X receptor, ASK1: apoptosis signaling kinase 1, Nrf2: nuclear factor-erythroid-2-related factor 2, CCR2/5: C-C chemokine receptor types 2 and 5, MRA: mineral corticoid receptor. Current status: * completed study, # ongoing study, $ discontinued study.

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
