# Peer review of "Common Drug Pipelines for the Treatment of Diabetic Nephropathy and Hepatopathy: Can We Kill Two Birds with One Stone?"

_ijms, 2020, doi:10.3390/ijms21144939_

Round 1

Reviewer 1 Report

The manuscript entitled “Common drug pipelines for the treatment of diabetic nephropathy 2 and hepatopathy: Can we Kill Two Birds with One Stone?” by Sumida Yoshio et al, is a review highlighting the common pathways involved in diabetic nephropathy and hepatopathy. Due to these molecular similarities, the Authors describe the possibility, based on preclinical animal studies and clinical trials, to use the same pharmacological treatment for both these diseases. The review reports an overview of the main drugs for the therapies and appears interesting to the readers because both these disorders represent a global health problem for the high incidence and prevalence. Moreover, these diseases can have a high impact on the quality life of patients. For these reasons, to identify a better therapy could be an important goal.

However, some points must be addressed.

Major points:

  1. The topic is very complex and the way how the Authors present the manuscript makes it hard to follow. Mainly in the paragraph 3, the description appears disorganized and non-homogeneous. I suggest to follow the same scheme in every section (3.1-3.10). The Authors should report in the text the same order of table 1, describing the molecule, the mechanism of action/pathway, the animal models and the clinical trials first in NAFLD/NASH with benefits for CDK and other way around. Finally, the quoted clinical trials should be organized as completed, in progress or discontinued.
  2. The animal models studies and the clinical trials mentioned across the manuscript (for example in section 3.3.1) should be described more in details so that the readers can appreciate the meaning and the relevance.
  3. The conclusion section is too concise, the authors have to implement it by including the challenges for the clinical management of the patients as well as a personal point of view of the more promising treatment.

Minor points:

  1. Table 1 is not mentioned in the text. Please add.

Furthermore, add lines to separate better the class of molecules (for example the chemical class of Liraglutide or Oltipraz are unclear).

The references quoted in table 1 are not consistent with the text (for example Saroglitazar is 36 in table and 59 in the text).

  1. The notation order of the references in the text is completely messed up. Please check.
  2. In Figure 1 and in Table 1 the captions miss. Please add.
  3. Check typo, for example:

-line 53 “diseased”

-line 75 “hepatis”

-line 119 “CVD”

-lines 213-214,216 some words are highlighted

-Line spacing and font size are not uniform.

Author Response

Reviewer 1

Major points:

  1. The topic is very complex and the way how the Authors present the manuscript makes it hard to follow. Mainly in the paragraph 3, the description appears disorganized and non-homogeneous. I suggest to follow the same scheme in every section (3.1-3.10). The Authors should report in the text the same order of table 1, describing the molecule, the mechanism of action/pathway, the animal models and the clinical trials first in NAFLD/NASH with benefits for CDK and other way around. Finally, the quoted clinical trials should be organized as completed, in progress or discontinued.

Answer: We revised the “section 3” according to your kind advise. We introduced the molecule, the mechanism of action/pathway, the animal models and the clinical trials first in NAFLD/NASH with benefits for CKD.

  1. The animal models studies and the clinical trials mentioned across the manuscript (for example in section 3.3.1) should be described more in details so that the readers can appreciate the meaning and the relevance.

Answer: We revised the” section 3” and table 1 according to your kind advise.

  1. The conclusion section is too concise, the authors have to implement it by including the challenges for the clinical management of the patients as well as a personal point of view of the more promising treatment.

Answer: In agreement with you, we revised conclusion section.

Minor points:

  1. Table 1 is not mentioned in the text. Please add.The references quoted in table 1 are not consistent with the text (for example Saroglitazar is 36 in table and 59 in the text).                                                         Answer:In agreement with you, we mentioned in the text about table 1. We added lined separate better the class of molecules. Incorrect references were revised.
  2. Furthermore, add lines to separate better the class of molecules (for example the chemical class of Liraglutide or Oltipraz are unclear).The notation order of the references in the text is completely messed up. Please check.                                                                Answer:Thank you for your kind advise, we checked notation order of the references.
  3. In Figure 1 and in Table 1 the captions miss. Please add.                                                                       Answer:we added the caption of Figure 1 and Table 1.
  4. Check typo, for example:                                        -line 75 “hepatis”  → “hepatic”                                  -lines 213-214,216 some words are highlighted → deleted                                                                      -Line spacing and font size are not uniform → Line spacing and font size are uniformed in revised manuscript.                                                                  -line 119 “CVD”    → “cardiovascular disease (CVD)” -line 53 “diseased”  → “disease”

Reviewer 2 Report

The authors address a topic of significant interest, in which there are considerable efforts in the therapeutic field, as demonstrated by the large amount of clinical studies reported. However, the review appears a little difficult to read, because the authors sometimes seem to limit themselves to do a list of clinical studies without introducing a more in-depth assessment of the putative efficacy of the proposed therapeutic approaches.

In particular, it would be useful to move many of the data into the table (study protocol numbers, patients involved, drug doses, etc ...) in order to make easier to read the text. The current table is confusing, and it does not allow to visualize the class to which the single drugs belong.

Furthermore, some pathogenetic mechanisms relevant for the therapeutic approaches are not sufficiently detailed. Also, in this regard it would be useful to draw a table summarizing the pathogenetic relevance of the different pathways common to NAFLD and CKD.

Author Response

Reviewer 2

The authors address a topic of significant interest, in which there are considerable efforts in the therapeutic field, as demonstrated by the large amount of clinical studies reported. However, the review appears a little difficult to read, because the authors sometimes seem to limit themselves to do a list of clinical studies without introducing a more in-depth assessment of the putative efficacy of the proposed therapeutic approaches.

In particular, it would be useful to move many of the data into the table (study protocol numbers, patients involved, drug doses, etc) in order to make easier to read the text. The current table is confusing, and it does not allow to visualize the class to which the single drugs belong. Furthermore, some pathogenetic mechanisms relevant for the therapeutic approaches are not sufficiently detailed. Also, in this regard it would be useful to draw a table summarizing the pathogenetic relevance of the different pathways common to NAFLD and CKD.

Answer: In agreement with you, we revised to make easier to read the text and reformed table 1.

Round 2

Reviewer 2 Report

The manuscript significantly improved. 

Author Response

As you suggested , We added severral papers. We performed English editing by your editing service.
